# Effect of geometric distortion correction on thickness and volume measurements of cortical parcellations in 3D T1w gradient echo sequences

**Christian Thaler** [1] *, **Jan Sedlacik**[2], **Nils D. Forkert**[3], **Jan-Patrick Stellmann**[4,5,6,7], **Gerhard Schön**[8], **Jens Fiehler**[1], **Susanne Gellißen**[1]

**1** Department of Diagnostic and Interventional Neuroradiology, University Medical Center Hamburg-Eppendorf, Hamburg, Germany, **2** Biomedical Engineering Department, School of Biomedical Engineering & Imaging Sciences, King's College London, London, United Kingdom, **3** Department of Radiology, University of Calgary, Calgary, Canada, **4** Department of Neurology, University Medical Center Hamburg-Eppendorf, Hamburg, Germany, **5** Institute for Neuroimmunology and Multiple Sclerosis, University Medical Center Hamburg-Eppendorf, Hamburg, Germany, **6** Department of Neuroradiology, APHM La Timone, CEMEREM, Marseille, France, **7** Aix-Marseille Univ, CNRS, CRMBM, UMR 7339, Marseille, France, **8** Institute of Medical Biometry and Epidemiology, University Medical Center Hamburg-Eppendorf, Hamburg, Germany

* c.thaler@uke.de

## Abstract

### Objective

Automated brain volumetric analysis based on high-resolution T1-weighted MRI datasets is a frequently used tool in neuroimaging for early detection, diagnosis, and monitoring of various neurological diseases. However, image distortions can corrupt and bias the analysis. The aim of this study was to explore the variability of brain volumetric analysis due to gradient distortions and to investigate the effect of distortion correction methods implemented on commercial scanners.

### Material and methods

36 healthy volunteers underwent brain imaging using a 3T magnetic resonance imaging (MRI) scanner, including a high-resolution 3D T1-weighted sequence. For all participants, each T1-weighted image was reconstructed directly on the vendor workstation with (DC) and without (nDC) distortion correction. For each participant's set of DC and nDC images, FreeSurfer was used for the determination of regional cortical thickness and volume.

### Results

Overall, significant differences were found in 12 cortical ROIs comparing the volumes of the DC and nDC data and in 19 cortical ROIs comparing the thickness of the DC and nDC data. The most pronounced differences for cortical thickness were found in the precentral gyrus, the lateral occipital and postcentral ROI (2.69, -2.91% and -2.79%, respectively) while cortical volumes differed most prominently in the paracentral, the pericalcarine and lateral occipital ROI (5.52%, -5.40% and -5.11%, respectively).

**Data Availability Statement:** All relevant data are within the paper and its Supporting Information files. The data can be found in Table 1.

**Funding:** The author(s) received no specific funding for this work.

**Competing interests:** The authors have declared that no competing interests exist.

## Conclusion

Correcting for gradient non-linearities can have significant influence on volumetric analysis of cortical thickness and volume. Since the distortion correction is an automatic feature of the MR scanner, it should be stated by each study that applies volumetric analysis which images were used.

## Introduction

Automated brain volumetry is a frequently used tool in neuroimaging for early detection, diagnosis, and monitoring of various neurological diseases, such as Alzheimer's disease, multiple sclerosis, epilepsy, etc. [1]. With a growing number of commercial volumetric tools the use of brain atrophy measures will become more frequent in clinical routine and will not be limited to clinical studies. An important requirement for brain volumetric analysis is a high sensitivity to small changes of global and focal brain volumes as well as a good reproducibility. Numerous technical factors can influence the results of brain volumetric analysis, such as acquisition protocols, intra- and interscanner variablilty, scanner system upgrades or patient movement [2, 3]. However, only some of these factors can be adjusted in single site studies, e.g. by using the same MR scanner, MR image sequence parameters, and software versions for the automatic brain volume analysis. Additionally, image distortion can influence the results of automated brain volumetry and need to be corrected to achieve good reproducibility and to allow accurate comparisons of quantitative results.

Image distortion in magnetic resonance imaging (MRI) has six potential sources: scale errors (linear) in gradient fields, shimming anomalies on the main magnet, chemical shift, B0 eddy currents, nonlinearities of gradient fields, and magnetic susceptibility variations in various anatomical structures [4]. The most prominent cause of image distortion in structural MRI are nonlinearities of gradient fields, which impact geometric and image intensity accuracy [5]. Previous studies have demonstrated that gradient distortions can influence the results of brain volumetric analysis and evaluated different methods for distortion corrections [5–8]. Correcting for gradient non-linearities improves image intensity reproducibility and can reduce volumetric errors caused by system variations. However, these methods demand in-depth expertise in the field and post-processing of the acquired images can be time consuming. Therefore, MRI vendors started to supply software for gradient corrections, which offer distortion corrected images generated directly on the vendor's workstation.

The effect of correcting image distortions on volumetric measures has been described before by numerous studies in great technical detail [5–8]. Those previous studies focused on the feasibility of their in-house developed distortion correction methods and did not report the impact of gradient non-linearities on volumetric measures. With the increasing availability of commercial software, volumetric analysis will be applied more frequently in clinical routine and research and clinicians as well as researchers should be aware of confounding factors such as gradient nonlinearities. Thus, the aim of the present work was to quantify the effect and magnitude of distortion corrections by utilizing a typical and practical scenario, i.e. volume measurements using the distortion correction methods provided by the MRI vendor. We hypothesized that the application of distortion correction has a significant and relevant impact on volumetric measurements. As technical studies have shown that distortion artifacts are lower in the isocenter and increase with distance, we further hypothesized that this might also be measurable in our data.

## Materials and methods

36 healthy volunteers (13 men, 23 women, aged 21 to 81 years, mean age 41.3 ± 13.4 years) underwent MR imaging at 3T (Skyra, Siemens Medical Solutions, Erlangen, Germany) between March 2013 and February 2015. The study was approved by the local Ethics Committee Hamburg (Ethik-Kommission der Ärztekammer Hamburg) following the guidelines of the Declaration of Helsinki and all subjects provided written informed consent.

Among others, the imaging protocol included a 3D Magnetization Prepared Rapid Acquisition Gradient Echo (MPRAGE) T1-weighted sequence with scan-matrix = 256x192x192, voxel size = 0.94x0.94x0.90mm$^3$, FOV = 240x180mm, slice thickness = 0.9mm, TR = 1900ms, TE = 3.4ms, and TI = 1100ms acquired in 5m55s. A 32 channel head coil was used and each patient was imaged with full brain coverage. No motion correction was applied.

For all participants, each T1-weighted dataset was generated directly on the Siemens workstation with distortion correction (DC) and without distortion correction (nDC). An example of the images with and without distortion correction is given in Fig 1. The distortion correction algorithm applied was the standard-algorithm implemented on the system as provided by the vendor (syngo MR E11, Siemens Medical Solutions, Erlangen, Germany). For further information about the 3D distortion correction algorithm please see *https://patents.google.com/patent/DE19540837A1/en*.

### FreeSurfer cortical thickness and volume

For each participant's set of DC and nDC images, the FreeSurfer software (version 7.1) for cortical reconstruction and volumetric segmentation (http://surfer.nmr.mgh.harvard.edu/) was used to determine regional cortical thickness and volume using FreeSurfer's recon_all pipeline. Freesurfer's segmentations and parcellations were inspected visually by an experienced neuroradiologist (eight years of experience) and corrected if necessary. Cortical thickness and volume measurements were quantified using the Desikan-Killiany-Tourville (DKT) atlas parcellation scheme and then compared for each ROI between DC and nDC data (Fig 2). The percentage difference was calculated using following formula: Difference thickness in percent

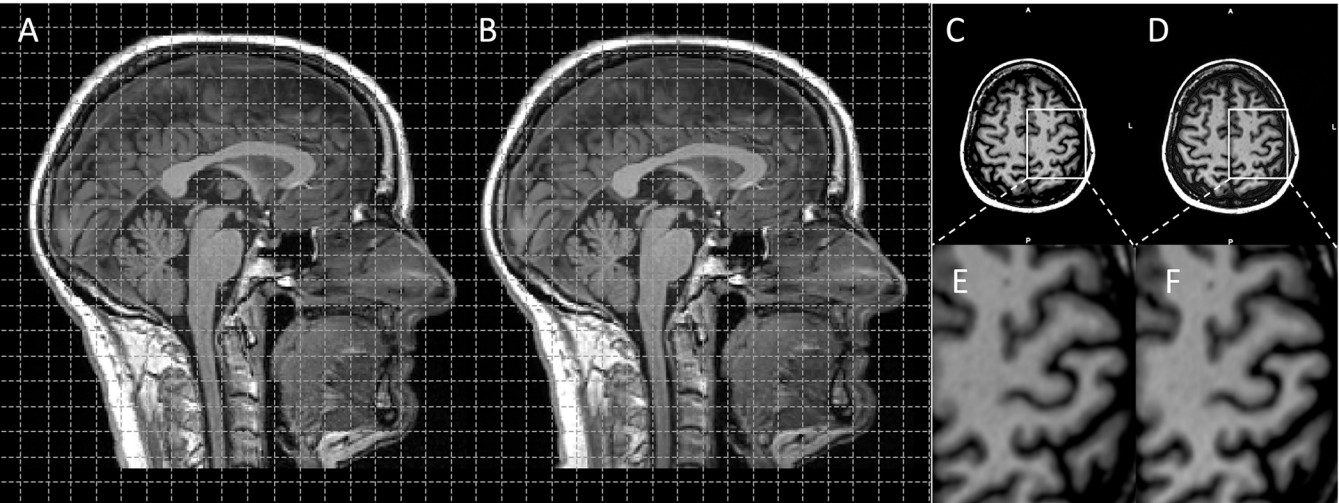

**Fig 1.** Example of T1w MPRAGE images in sagittal and axial plane with (A and C) and without (B and D) distortion correction. The distortion effects were most prominent in the central region but difficult to detect visually (E and F).

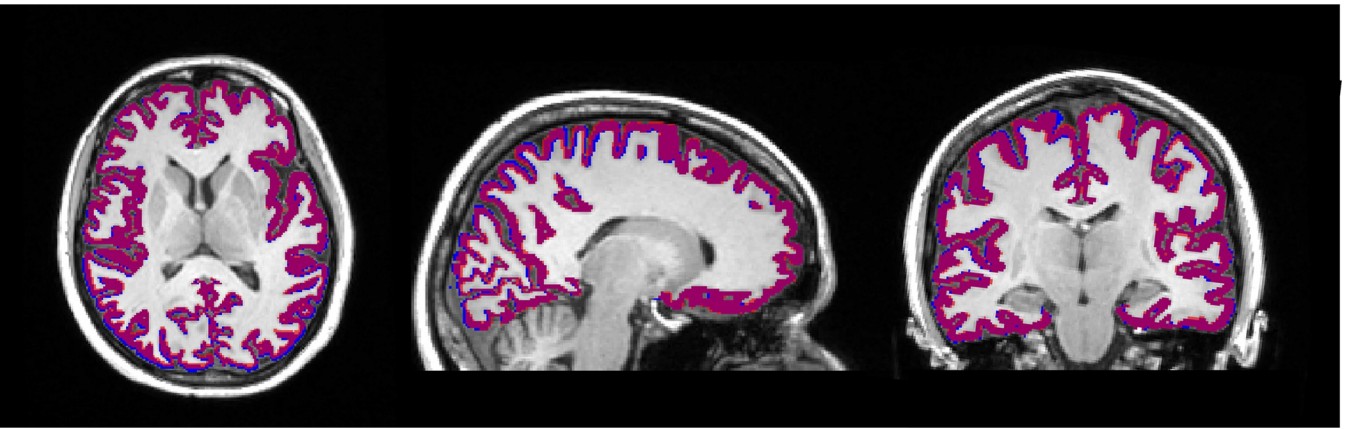

**Fig 2. Exemplary overlay of the cortical segmentations from the FreeSurfer parcellations of the distortion corrected (DC) (blue) and not distortion corrected (nDC) (red) images in axial, coronal, and sagittal view.** Purple areas indicate the overlap between the two parcellations.

= (DC(thickness)-nDC(thickness))/nDC(thickness) and Difference volume in percent = (DC (volume)-nDC(volume))/nDC(volume).

Isocenter coordinates for each patient were derived from the origin of the anatomical coordinate system that were transferred to the respective image coordinate system. The isocenter was estimated applying the corresponding FreeSurfer utilities based on the information stored in the DICOM header. This was necessary since the recon-all stream transforms the original data into a FreeSurfer space and changes the coordinates of the magnetic isocenter alike. After determining the coordinates of the magnetic isocenter in the FreeSurfer space we calculated the distance for each FreeSurfer parcel (using the center of gravity of each ROI) from the magnetic isocenter.

## Statistics

Statistical analysis was performed using Statistics in R 3.0.0 and IBM SPSS 27.0. A paired t-test was used to compare the cortical thickness and volume measurements between the DC and nDC images. The false discovery rate (FDR) was used to correct for multiple comparisons. A p-value <0.05 was considered statistically significant.

Concordance correlation coefficients (CCC) were computed to assess the inter-method reliability of brain morphometry results obtained from the DC and nDC images. Furthermore, we computed a linear mixed model, using the command *lmer* in R, with ROI thickness, volume, and distance to the magnetic isocenter along each axis (in mm) as fixed effects and the percentage difference in thickness and volume between the DC and nDC data as dependent variables separately, with patient ID entered as random effect.

## Results

### Cortical thickness and volume

Of the 31 cortical ROIs per hemisphere in each patient, 0.76% of the regions remained equal when comparing thickness measures derived from DC and nDC data. 54.75% of the ROIs showed decreased thickness measures in the nDC data compared to the DC data and 44.49% a positive difference. For volume measurements, 0.22% of the regions remained equal, while 59.59% of the ROIs showed decreased volume measurements in the nDC data compared to the DC data and 40.19% regions a positive difference. Overall, significant differences were

**Table 1. Mean values, standard deviations (in parentheses), and 95% confidence intervals for volume and thickness for each cortical ROI.**

| | Thickness in mm | | | | Volume in ml | | | |
|---|---|---|---|---|---|---|---|---|
| | DC | nDC | 95% CI | p | DC | nDC | 95% CI | p |
| Caudal anterior cingulate | **2.428 (0.176)** | **2.448 (0.174)** | **-0.033 --0.006** | **0.010** | 2.781 (0.869) | 2.810 (0.808) | -0.088–0.029 | 0.485 |
| Caudal middle frontal | **2.529 (0.154)** | **2.548 (0.134)** | **-0.032 - -0.007** | **0.007** | 6.667 (1.514) | 6.623 (1.161) | -0.216–0.303 | 0.816 |
| Cuneus | **1.800 (0.106)** | **1.772 (0.114)** | **0.016–0.040** | **<0.001** | **4.184 (0.748)** | **4.027 (0.758)** | **0.104–0.211** | **<0.001** |
| Entorhinal | 3.283 (0.238) | 3.297 (0.224) | -0.036–0.009 | 0.296 | 1.688 (0.355) | 1.695 (0.333) | -0.038–0.024 | 0.746 |
| Fusiform | **2.675 (0.125)** | **2.661 (0.120)** | **0.005–0.024** | **0.009** | 8.891 (1.474) | 8.791 (1.440) | -0.001–0.202 | 0.115 |
| Inferior parietal | **2.460 (0.120)** | **2.410 (0.126)** | **0.041–0.059** | **<0.001** | **14.142 (2.597)** | **13.523 (2.418)** | **0.459–0.779** | **<0.001** |
| Inferior temporal | 2.920 (0.129) | 2.917 (0.123) | -0.005–0.012 | 0.537 | 13.419 (1.793) | 13.380 (1.678) | -0.079–0.159 | 0.655 |
| Insula | 3.083 (0.143) | 3.092 (0.146) | -0.022–0.003 | 0.182 | 6.301 (0.755) | 6.319 (0.732) | -0.057–0.020 | 0.492 |
| Isthmus cingulate | 2.292 (0.139) | 2.284 (0.152) | -.006–0.022 | 0.311 | 2.620 (0.474) | 2.617 (0.515) | -0.042–0.048 | 0.893 |
| Lateral occipital | **2.139 (0.101)** | **2.079 (0.101)** | **0.051–0.069** | **<0.001** | **13.385 (2.096)** | **12.757 (2.049)** | **0.489–0.767** | **<0.001** |
| Lateral orbito-frontal | 2.615 (0.121) | 2.621 (0.122) | -0.016–0.003 | 0.225 | 8.995 (1.067) | 8.980 (1.079) | -0.044–0.074 | 0.737 |
| Lingual | **1.956 (0.132)** | **1.935 (0.132)** | **0.006–0.036** | **0.016** | **7.274 (1.465)** | **7.137 (1.450)** | **0.025–0.248** | **0.044** |
| Medial orbito-frontal | 2.450 (0.129) | 2.448 (0.144) | -0.011–0.016 | 0.755 | 4.960 (0.635) | 4.945 (0.639) | -0.020–0.051 | 0.532 |
| Middle temporal | **2.845 (0.126)** | **2.826 (0.125)** | **0.011–0.026** | **<0.001** | **15.133 (2.494)** | **14.800 (2.387)** | **0.230–0.437** | **<0.001** |
| Paracentral | **2.359 (0.180)** | **2.407 (0.174)** | **-0.068 - -0.028** | **<0.001** | **3.899 (0.575)** | **4.131 (0.549)** | **-0.306 - -0.158** | **<0.001** |
| Parahippo-campal | 2.717 (0.211) | 2.720 (0.206) | -0.015–0.009 | 0.699 | 2.261 (0.301) | 2.253 (0.309) | -0.023–0.389 | 0.749 |
| Pars opercularis | 2.600 (0.162) | 2.604 (0.157) | -0.015–0.008 | 0.618 | **4.601 (0.954)** | **4.504 (0.919)** | **0.020–0.175** | **0.039** |
| Pars orbitalis | 2.675 (0.172) | 2.687 (0.178) | -0.025–0.001 | 0.103 | 2.331 (0.381) | 2.314 (0.367) | -0.008–0.040 | 0.325 |
| Pars triangularis | 2.419 (0.158) | 2.414 (0.151) | -.006–0.017 | 0.402 | 4.406 (0.936) | 4.347 (0.874) | -0.016–0.135 | 0.238 |
| Pericalcarine | **1.462 (0.113)** | **1.437 (0.118)** | **0.015–0.035** | **<0.001** | **2.019 (0.451)** | **1.921 (.443)** | **0.075–0.121** | **<0.001** |
| Postcentral | **2.087 (0.128)** | **2.033 (0.105)** | **0.028–0.081** | **<0.001** | 10.436 (1.757) | 10.420 (1.604) | -0.113–0.146 | 0.853 |
| Posterior cingulate | **2.321 (0.104)** | **2.334 (0.109)** | **-0.026- -0.001** | **0.047** | **3.361 (0.557)** | **3.466 (0.547)** | **-0.152 - -0.059** | **<0.001** |
| Precentral | **2.510 (0.223)** | **2.580 (0.156)** | **-0.109- -0.030** | **0.002** | **12.747 (1.988)** | **13.133 (1.649)** | **-0.644 - -0.129** | **0.016** |
| Precuneus | **2.314 (0.133)** | **2.291 (0.137)** | **0.010–0.036** | **0.002** | 9.970 (1.717) | 9.873 (1.509) | -0.115–0.309 | 0.516 |
| Rostral anterior-cingulate | **2.609 (0.150)** | **2.625 (0.144)** | **-0.028- -0.004** | **0.016** | 3.087 (0.847) | 3.118 (0.849) | -0.061–0.001 | 0.103 |
| Rostral middle frontal | 2.366 (0.118) | 2.367 (0.117) | -0.009- -0.008 | 0.866 | 12.019 (0.236) | 11.914 (2.136) | -0.078–0.289 | 0.439 |
| Superior frontal | **2.648 (0.135)** | **2.685 (0.120)** | **-0.048- -0.025** | **<0.001** | **26.292 (4.179)** | **26.773 (3.730)** | **-0.839 - -0.122** | **0.009** |
| Superior parietal | **2.147 (0.130)** | **2.112 (0.124)** | **0.025–0.044** | **<0.001** | 10.696 (2.006) | 10.718 (1.634) | -0.336–0.293 | 0.922 |
| Superior temporal | **2.913 (0.136)** | **2.903 (0.142)** | **0.002–0.018** | **0.024** | **17.234 (2.262)** | **17.066 (2.324)** | **0.047–0.290** | **0.024** |
| Supramarginal | **2.546 (0.126)** | **2.528 (0.135)** | **0.007–0.029** | **0.005** | 10.223 (2.202) | 10.388 (1.884) | -0.469–0.139 | 0.462 |
| Transverse temporal | 2.312 (0.274) | 2.293 (0.273) | 0.000–0.038 | 0.078 | 1.032 (0.265) | 1.020 (0.267) | -0.003–0.026 | 0.238 |

A paired t-test was used to compare the results from the distortion corrected (DC) and not distortion corrected (nDC) images. Significant differences (p < 0.05) are highlighted in bold. The false discovery rate (FDR) was used to correct for multiple comparisons.

found for 12 out of 31 cortical ROIs regarding the volume and in 19 cortical ROIs with respect to the thickness. An overview of mean volumes and thickness for each cortical ROI is displayed in Table 1.

Thickness measurement differences between DC and nDC data were most pronounced in the precentral and paracentral ROIs (mean percentage difference 2.69% and 1.96%) with increased thickness measures in the nDC images, and most pronounced in the lateral occipital and postcentral ROIs (-2.91% and -2.79%) with decreased thickness measures in the nDC images. Smallest deviations of thickness measures were found in the rostral middle frontal and parahippocampal ROI (0.02% and 0.1%). Fig 3 displays the percentage differences for thickness comparing DC and nDC images.

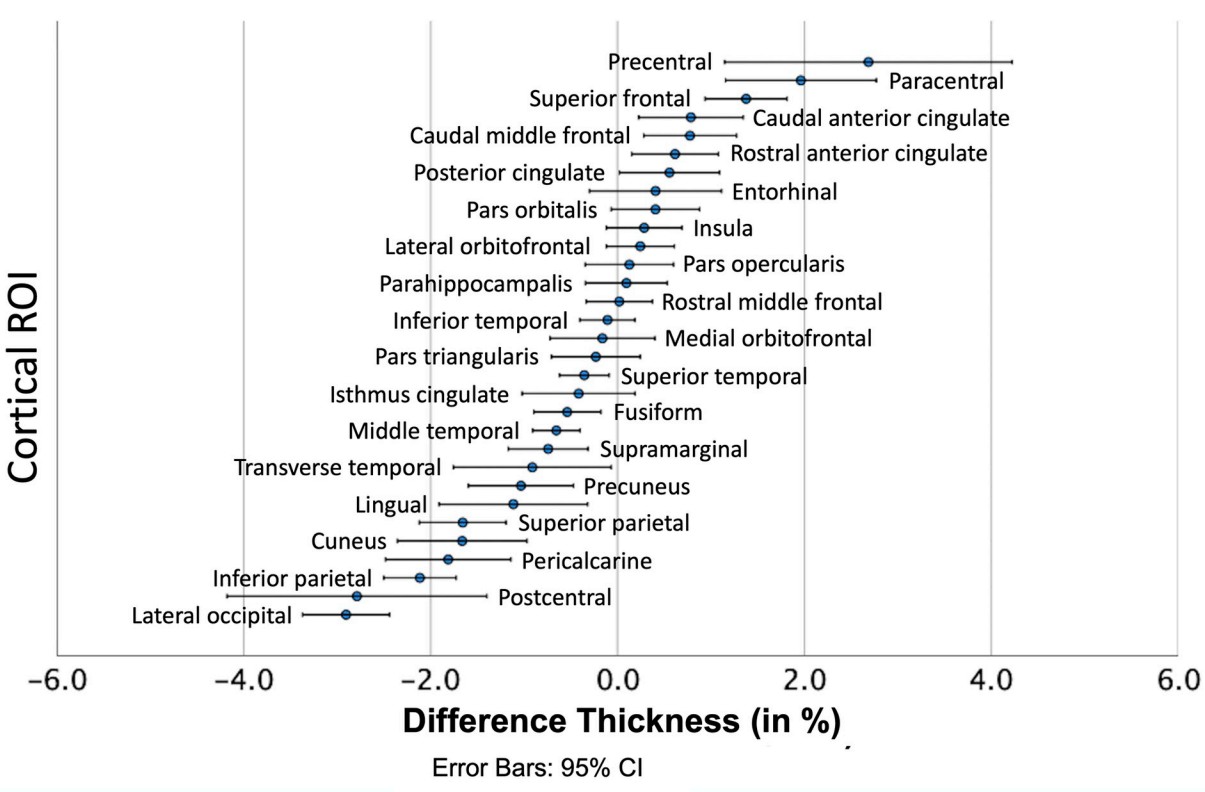

**Fig 3. Mean differences of thickness for each ROI between nDC and DC data.** The ROIs (y axis) are sorted by their mean percentage differences. The dot indicates the mean, and the error bars indicate the 95% confidence interval.

Volume measurement differences between DC and nDC data were most pronounced in the paracentral and precentral ROIs (mean percentage difference 5.52% and 3.00%) with increased volume measurements in the nDC images, and most pronounced in the pericalcarine and lateral occipital ROIs (-5.40% and -5.11%) with decreased thickness measurements in the nDC images. Smallest deviations of volume measures were found in the superior parietal and postcentral ROI (0.01% and -0.05%). Fig 4 displays the percentage differences for volume comparing DC and nDC images.

The overall concordance correlation coefficient (CCC) for thickness comparing the DC and nDC measurements was 0.985. The overall CCC for volume measurements comparing the DC and nDC data was 0.995. Fig 5 and S1 Table display the CCC for each cortical ROI comparing DC and nDC measurements.

## Linear mixed model analysis

The linear mixed model analyses predicting percentage differences for thickness and volume between DC and nDC images are summarized in Table 2. For percentage differences in thickness, significant predictors were the distance to the magnetic isocenter in y and z axis. While the distance to the magnetic isocenter in z axis was positively associated with percentage differences in thickness, the distance to the magnetic isocenter in y axis was negatively associated.

For percentage differences in volume, significant predictors were the distance to the magnetic isocenter in y and z axis but also ROI volume and thickness. While the distance to the magnetic isocenter in z axis and ROI thickness were positively associated with percentage

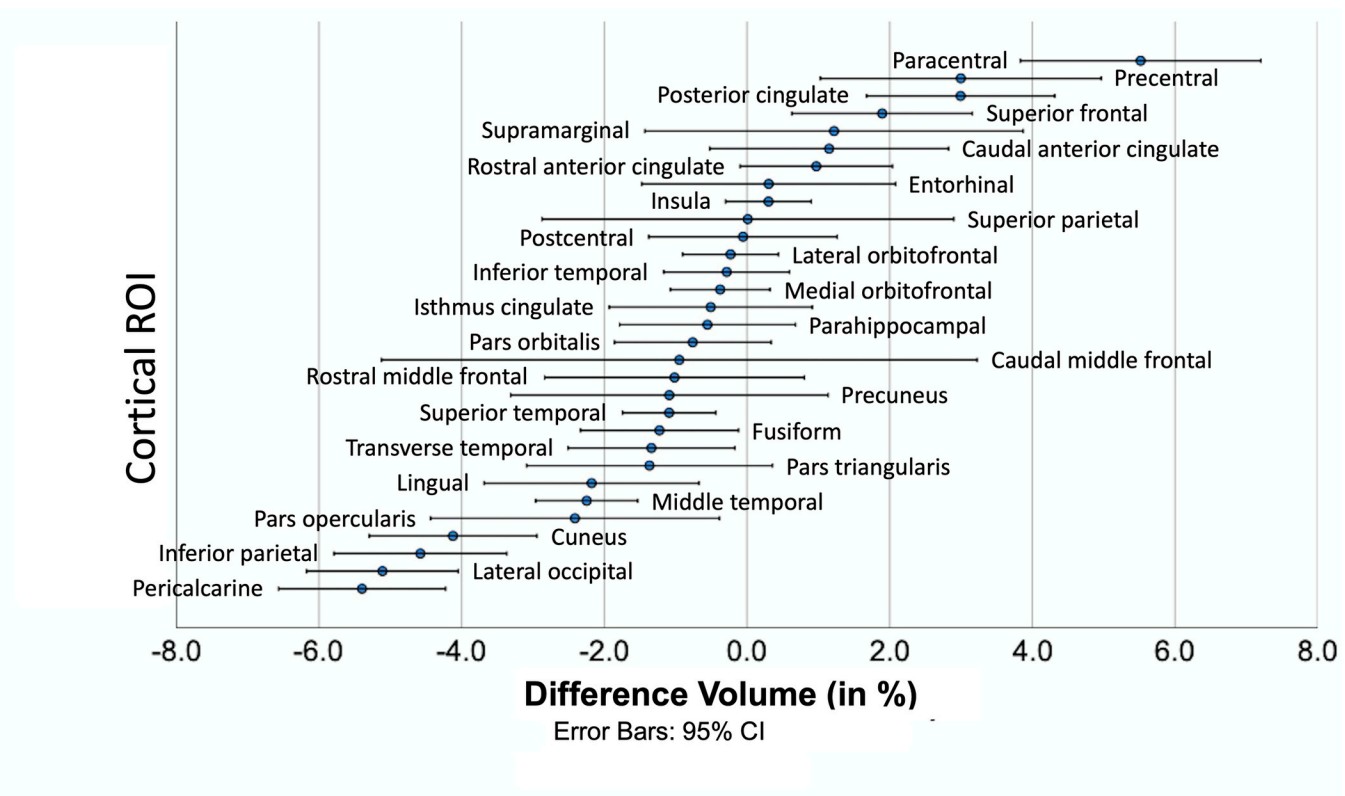

**Fig 4. Mean differences of volume for each ROI between nDC and DC data.** The ROIs (y axis) are sorted by their mean percentage differences. The dot indicates the mean and the error bars indicate the 95% confidence interval.

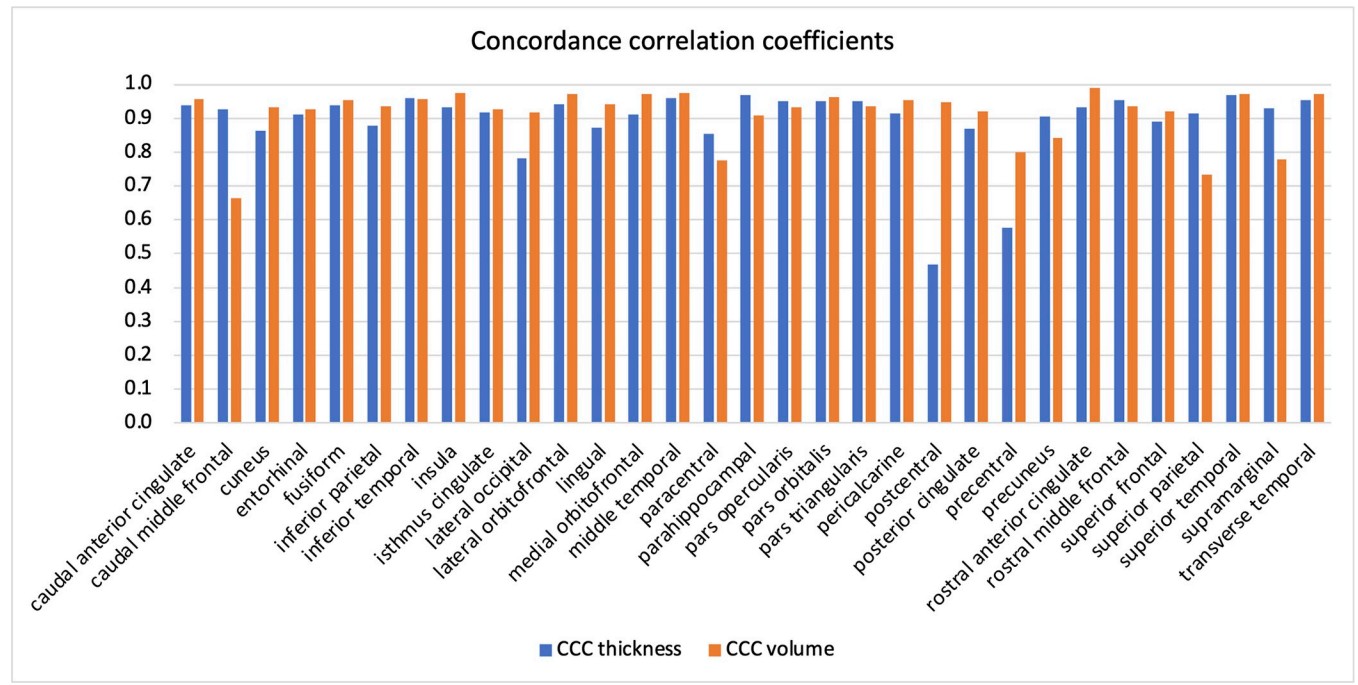

**Fig 5. Bar graph displaying the concordance correlation coefficients between DC and nDC data for each cortical ROI.**

**Table 2. Results of the multiple linear regression analysis.**

| | Percentage difference in thickness | | | Percentage difference in volume | | |
|---|---|---|---|---|---|---|
| | Estimate | CI | p | Estimate | CI | p |
| Distance to isocenter (x)(mm) | 8.93e-04 | -0.003–0.005 | 0.629 | 9.43e-04 | -0.01–0.008 | 0.843 |
| Distance to isocenter (y)(mm) | **-0.012** | **-0.017 –-0.007** | **<0.001** | **-0.045** | **-0.058 –-0.031** | **<0.001** |
| Distance to isocenter (z)(mm) | **0.019** | **0.015–0.022** | **<0.001** | **0.015** | **0.015–0.024** | **0.001** |
| ROI Volume (ml) | -0.019 | -0.039–0.001 | 0.066 | **-0.129** | **-0.181 –-0.076** | **<0.001** |
| ROI Thickness (mm) | 0.082 | -0.289–0.449 | 0.665 | **1.612** | **0.642–2.563** | **<0.001** |

differences in volume, the distance to the magnetic isocenter in y axis and ROI volume were negatively associated. The association between distance from the magnetic isocenter along each axis and percentage difference in thickness and volume are displayed in S1 and S2 Figs.

## Discussion

Brain morphometric analyses are frequently applied in various neurological conditions to detect pathological brain atrophy patterns but also as a surrogate marker to evaluate disease progression and improve clinical-radiological correlations [9, 10]. A high sensitivity to small changes of global and focal brain volumes as well as a good reproducibility are essential to compare and evaluate patient data in cross-sectional and longitudinal studies. Among other factors, image distortion can influence the results significantly. In this study, we evaluated the effect of gradient non-linearities on computational brain morphometry analyses. More precisely, we compared the results of automated brain morphometry between nDC and DC images as supplied by the commercial software implemented in Siemens RT Dot engine.

By comparing the results of brain morphometry obtained from nDC and DC images, we found significant differences in cortical thickness and volume in more than half of the cortical ROIs. The most pronounced differences for cortical thickness were found in the precentral gyrus, the lateral occipital, and postcentral ROIs (2.69, -2.91% and -2.79%, respectively) while cortical volumes differed most prominently in the paracentral, the pericalcarine, and lateral occipital ROIs (5.52%, -5.40% and -5.11%, respectively). These differences are relatively high and can have a huge impact when evaluating brain atrophy rates in cross-sectional and longitudinal studies. For example, Hardmeier et al. documented an annual brain tissue loss of 0.5–1% in a multiple sclerosis study cohort, compared to 0.1–0.3% in a healthy control cohort [11]. Also, when just evaluating the annual loss of grey matter, the annual mean decrease of grey matter volume in multiple sclerosis patients has been reported to be 3.3% and 1.1% in healthy controls [12]. Furthermore, numerous studies have found regional differences for cortical atrophy patterns in multiple sclerosis patients compared to healthy controls, *i.e.* in early disease stages predominant atrophy of the gray matter in the fronto-temporal areas such as the superior temporal gyrus, the superior and middle frontal gyri, and the motor cortex [13, 14]. In more detail, Narayana et al. reported significant differences in cortical thickness between multiple sclerosis patients and an age- and sex-matched control group in numerous cortical regions, with differences ranging from 0.15% to 10.74% [15].

Volumetric measures are also used for diagnostic purposes and evaluating disease progression in other neurological conditions, such as dementia, Parkinson disease, or traumatic brain injury [16–18]. For example, in Alzheimer's disease, cortical volume and thickness features have been examined to differentiate between subtypes of the disease [19]. Hence, distortion effects leading to in- or decreasing volume and thickness measures can affect the results significantly. With the growing popularity of commercial volumetric tools volumetric analysis will

be conducted on a more regular basis in clinical routine. Therefore, clinicians and researchers must be aware of the effects caused by distortions on volumetric results, especially when using cut-off values to differentiate between healthy aging and affected patients.

However, none of the aforementioned studies reported whether scans with or without DC were used for their analyses. The reported differences in volumes and cortical thickness between patient groups are thereby of the same magnitude as the differences we were able to detect when comparing nDC and DC measurements. Including DC and nDC datasets in an analysis can therefore cause a systemic bias in cross-sectional and longitudinal studies. Of course, this bias is more likely in multi-site studies assuming that one site uses the distortion correction provided by the vendor and the others do not. Furthermore, McRobbie et al. demonstrated that the effect of gradient non-linearities on image distortion is also system dependent [20]. Applying distortion corrections can decrease system dependency, though it does not eliminate it completely [7]. The scans in our study were all acquired on the same MR scanner. Therefore, we were not able to investigate the system dependency of distortions between vendors in detail. Since the DC and nDC images are both simultaneously generated by default and both images could be used for brain volumetric analysis, the effect of distortion correction is also important for single-center studies. Therefore, we encourage authors to be aware of this effect and clearly describe in their methods whether DC has been applied to prevent this unwanted bias.

Our results show most pronounced differences for cortical volume and thickness in the frontal, occipital, and parietal lobes, while differences in the temporal lobes were rather subtle. Moreover, CCC for volume and thickness measurements were lowest for the cortical ROIs of the frontal and parietal regions. A possible explanation for this finding might be that gradient linearity decreases significantly with increasing distance from the magnetic isocenter, which is pronounced along the z-axis [6]. Our findings support this with distance to the magnetic isocenter along the z-axis as positive predictor for thickness and volume differences between the DC and nDC images. Interestingly, the distance from the magnetic isocenter along the y-axis also showed a significant but negative association with the percentage difference of volume and thickness. It was reported by Jovicich et al. that head positioning can have an impact on distortion effects [5]. Therefore, even in single-center studies, i.e., using the same MR system with standardized imaging protocols and using the same software version, gradient non-linearities can have inhomogeneous distortion effects, leading to shrinking or inflation of different cortical areas, depending on the head position. However, this was not formally investigated in this presented study and remains speculative.

There are several limitations of this study that need to be mentioned. First, we only used the open-source FreeSurfer software to investigate the effect of distortion correction on brain volumetry and cannot make any conclusions regarding this effect when using different brain morphometry software. However, we are unaware of commercially available software that includes correction of gradient non-linearities. Therefore, we are confident that the impact of gradient non-linearities will be similar in other software versions. Second, we only included healthy volunteers in this study. Therefore, we cannot determine the effect of distortion correction on brain volumetry with gray or white matter atrophy or disease associated lesions. As mentioned above, we acquired our images on a single MR scanner and cannot investigate the system dependency of distortions in detail. Also, we used a 3D correction method and cannot our results with studies using a 2D correction method which are still in use by many vendors. However, the advantage of 3D distortion correction methods is their ability to provide more accurate and precise correction of image distortions, leading to improved image quality, better diagnostic accuracy [21]. In addition, there are other sources for distortions in brain MRI that can influence volumetric analysis but were not included in our study. For the linear mixed

model we determined p values using ANOVA. We understand that determining p values can be controversial for linear mixed models and it is often recommended to omit them.

In conclusion, our results show that correcting for gradient non-linearities can have significant influence on volumetric analysis of cortical thickness and volume. Since the distortion correction is an automatic feature of the MR scanner, it should be described in detail by each study that applies volumetric analysis which images were used.

## Supporting information

**S1 Fig. Association between the distance from the magnetic isocenter and the volume difference between the DC and nDC cortical parcellations.** Values are displayed for each axis (x,y,z) in separate colors.
(TIF)

**S2 Fig. Association between the distance from the magnetic isocenter and the thickness difference between the DC and nDC cortical parcellations.** Values are displayed for each axis (x,y,z) in separate colors.
(TIF)

**S1 Table. Concordance correlation coefficients between DC and nDC data for each cortical ROI including upper and lower 95% confidence intervals.**
(DOCX)

## Author Contributions

**Conceptualization:** Jan Sedlacik, Susanne Gellißen.

**Data curation:** Christian Thaler, Nils D. Forkert, Susanne Gellißen.

**Formal analysis:** Christian Thaler, Gerhard Schön, Susanne Gellißen.

**Funding acquisition:** Jens Fiehler.

**Investigation:** Christian Thaler, Gerhard Schön, Susanne Gellißen.

**Methodology:** Christian Thaler, Jan Sedlacik, Nils D. Forkert, Susanne Gellißen.

**Project administration:** Susanne Gellißen.

**Resources:** Jan-Patrick Stellmann, Jens Fiehler.

**Supervision:** Christian Thaler, Jan Sedlacik, Nils D. Forkert, Jens Fiehler, Susanne Gellißen.

**Validation:** Christian Thaler, Jan Sedlacik, Nils D. Forkert, Jan-Patrick Stellmann.

**Visualization:** Christian Thaler.

**Writing – original draft:** Christian Thaler.

**Writing – review & editing:** Jan Sedlacik, Nils D. Forkert, Jan-Patrick Stellmann, Gerhard Schön, Jens Fiehler, Susanne Gellißen.

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
