## [Decision Letter · Decision Letter 0]

19 Sep 2022

PONE-D-22-16057Effect of Geometric Distortion Correction on Thickness and Volume Measurements of Cortical Parcellations in 3D T1w gradient echo sequencesPLOS ONE

Dear Dr. Thaler,

Thank you for submitting your manuscript to PLOS ONE. After careful consideration, we feel that it has merit but does not fully meet PLOS ONE’s publication criteria as it currently stands. Therefore, we invite you to submit a revised version of the manuscript that addresses the points raised during the review process. In particular, Reviewer 2 raised significant concerns about the lack of ground truth in some experiments, a lack of details in the methods and results, and the overall impact of the results. Certainly, transparency and consistency of procedures are cornerstones of science, but beyond reinforcing these notions, the reviewer wondered if additional recommendations regarding best practices in applying distortion correction in imaging studies could be derived from the results. 

We look forward to receiving your revised manuscript.

Kind regards,

Dzung Pham

Academic Editor

PLOS ONE

Journal Requirements:

Reviewers' comments:

Reviewer's Responses to Questions

**Comments to the Author**

1. Is the manuscript technically sound, and do the data support the conclusions?

Reviewer #1: Yes

Reviewer #2: No

2. Has the statistical analysis been performed appropriately and rigorously? 

Reviewer #1: Yes

Reviewer #2: No

3. Have the authors made all data underlying the findings in their manuscript fully available?

Reviewer #1: Yes

Reviewer #2: Yes

4. Is the manuscript presented in an intelligible fashion and written in standard English?

Reviewer #1: Yes

Reviewer #2: Yes

5. Review Comments to the Author

Reviewer #1: Well written paper, with some minor English issues (see below)

Not very innovative, standar statistics and existing public software tools were used for evaluation of the effects of different methods for distorion correction. You could describe some the non-linearity models. No mathematical formalisms are presented, just mentioning and citing known analysis techniques.

Please enhance the quality of figures 2 and 3; the text is very blurred; I also recommend maximizing contrast by using black for the font.

Some corrections:

Page 2, line 22

lateraloccipital and postcentral

lateral occipital and postcentral

THIS IS PREFERRED, "postcentral" is fine

Please make this correction in all the document

Page 15, line 18

Of note, it was reported by Jovicich

? what do you mean by "Of note"? ask help form an English.spoken person

page 17 line 13

captured in or study

included in our study <- you refer to sources of distortion, the use of "capture" is incorrect.

Reviewer #2: This paper looks at the effect of gradient non-linearities corrections on 3D T1-w magnetic resonance (MR) images. Specifically, the paper compares the estimates from the output of the Freesurfer processing pipeline such as brain volumes and regional cortical thickness from images with and without correction of non-linearities (DC, as per authors). There is a lot of work in the paper, and I found the figures to be essential to the understanding of the manuscript.

Introduction

I found the introduction lacking fundamental details. The literature has significant contributions trying to address gradient non-linearities and their impact to the reconstructed images. They present the source, the physics behind their correction and potential improvements while their limitations. The authors site only three papers that address image distortions corrections. However, these only describe some of the problems including multi-site data and these are not included in the introduction. I recommend expanding to add a short review of others’ findings.

The authors state that numerous factors can affect the images. A little naming of them is needed in the introduction. Furthermore, the authors indicate that these factors can be easily adjusted in a single site study. I don’t believe the statement is true. There are factors that are not related to the technology used, unavoidable changes to the technology, errors from the coils, factors associated with the participant, the physiology (such as flow, susceptibility), and positioning in the scanner. What are the inherent errors of perfect data? Multiple of these factors are not able to be corrected. Rudrapatna et al. demonstrated that motion influences the non-linearities corrections. I believe some of these should be introduce in the introduction

The paper tests the use versus the non-use of a motion correction technique. However, which of these two approaches is correct or more accurate is not known. There is no ground true when scanning humans. Have the authors considered using phantom and what are the limitations of such approach? What is the novelty of this technique compared to others. There are multiple correction techniques, what differentiates this DC algorithm from others? There is a need to explain the correction technique, the expectations of the correction and the therefore elaborate a hypothesis. What problem is this paper really targeting? The hypothesis should be quantifiable and should state the reasoning to the change of using the DC technique. A significant change does not really indicate that there is a cause-consequence that drives a hypothesis.

Materials

Brain morphology is dependent on the age and gender of the participants. The demographics of this study are highly variable. A more detailed information will help understand the changes seen in the results. The study is based on one scanner, one software version, one protocol version, one correction technique. How long did it take to get the 36 subjects scanned? There are intricate changes to the technology that can biased the results here presented. Did anything else change? More detailed information is needed.

The methods sections should have enough information to allow others to reproduce the experiment. This is a requirement for transparency and reproducibility of any scientific publication. I think there are many sections that can be improved, and more information is needed.

• On the DC technique to further assess the results. This include the algorithm itself and the software version and others.

• The acquisition process, not just only the T1-w. length to acquire 36 participants, coil, landmarking, motion correction, etc.

• Freesurfer v7.1 is known to provide inaccurate parcellations of the brain, compared to previous versions. It often overestimates the temporal and parietal cortex. This refers to the robustness of the Freesurfer pipeline. Have the authors encounter these issues? Was there any mitigation action to correct the template transformations? Did the pipeline behave similarly between the nDC and the DC corrected data? Did all the transformations work accurately? How was this evaluated?

• The estimate of the isocentre needs an explanation. What was the purpose or intent for its estimate. The paragraph as listed does not indicate the intention of the estimate use.

• CCC is considered similar to ICC. How was it calculated, there are multiple ICC estimates depending on the data (see Koo 2016). When reporting ICC, the confidence intervals should also be reported. These determine if the ICC is excellent, good, or poor.

• LME model: given that the hypothesis is measuring difference, I understand the model used to fit the difference between the 2 images. However, where all the fixed effect used in the model simultaneously or one at the time? The explanation needs more information on the purpose of the modeling. The LME was not used to model the data, but it was used to measure factors that could influence the difference in the estimates. Results indicate a somewhow different use of the model. How was the p -value estimated? Was it a simple ANOVA between 2 different model? Which factors were used and why? If you use volume and thickness, are you expecting edge voxels to influence the results? In addition, given the variability in the demographics of the cohort used in the study, why were age and gender not used as factors as well?

Results:

Initially I found the results interesting and was surprised to see the region dependency of the estimates. It can be expected from the algorithm, and its special dependency in the correction technique. However, physiological variability can also lead to distortions that the algorithm can either not be able to or inaccurate try to correct. This is dependent of the DC algorithm.

As I looked further, I found some confusing information

• I recognized this is a small cohort. There is no information on how the difference, or the estimates were distributed across participants. Are these normally distributed? If not, is the t-test and LME model appropriate or is there a need for non-parametric test?

• Figure 2 and 3: they report % change. Percentage normalized to which value? I think the formula of the difference % [nDC-DC]/??? should be present in the axis to help their interpretation. This normalization can also bias the results. It is currently sorted based in most negative change to most positive change. When sorted based on anatomical region, were other information that could be inferred? Like most regions in the frontal lobe have statistically significant changes based on the t-test and why? What drives the changes in some regions compared to others.

• Table 1: If the 95% CI for the difference and therefore associated with Figue 2 and 3? For example, Caudalmiddlefrontal, it is not statistically significance different (t-test) in the volume, however, the variability in the difference estimates is high and crosses zero. Are there outliers, a particular participant driving this variability, is it a cohort variability? Further explanation is needed. The text speaks of summary values, which are usually good as a start point, but needs further detailed description of some examples.

• Figure 4, what drive the low CCC values? Is it the within variability of the measures?

Discussion

The discussion section describes the need to report the type of data (nDC or DC) used in the estimates to infer changes. I agree with the authors that this is relevant and important and should always be described in a paper. Again, this aligns with the importance of transparency and reproducibility of published data. However, in cross-sectional, particularly at a single centre, if all data are treated in the same manner, the biases due to non-linear gradients can be considered similar to all participants if collected within a short time frame. Furthermore, you might not be affected by the correction, considering that this is not better than non-corrected data, if not applied at all. However, the biases due to anatomical variability and others are still present. In longitudinal data there are other sources of errors that can affect your measure more than non-linear gradients.

The authors highlight the importance of the changed seen and compared those to anatomical changes over time in different pathologies reported in the literature. In the review, caution should be considered and group results by similarly managed data. Each study had chosen a processing pipeline and sometimes papers cannot be compared even less grouped together. I agree with the authors that head positioning is an important and unavoidable factor. How can the effect of this be minimized?

I find it difficult to relate the discussion to the proposed hypothesis. Can the hypothesis really be proven if not all regions are affected similarly? Are there edge-effects or outliers in the ROIs used? If so, how was this evaluated and corrected for?

How does the physiology affect the DC algorithm?

Minor comments

• Use of acronyms such as MR and MRI, while known, should always be introduced. Same for DC.

• Introduction: line 8 should be single site or single centre, and not single-side.

• Introduction: Page 5 line 1, confusing, needs references and further explanation.

• Figure1: please overlay with some transparency, hard to see the differences.

• Discussion: page 14 line 09 I would use dementia, as it is a spectrum. Also, it is Parkinson’s disease and there is a full stop after it that should be a comma.

• Discussion: page 14 line 22: This can lead… is repeated from 2 sentences before.

• Discussion: page 15 line 22 should say cortical areas, not areals

6. PLOS authors have the option to publish the peer review history of their article (what does this mean?). If published, this will include your full peer review and any attached files.

Reviewer #1: No

Reviewer #2: No

---

## [Author Response · Author response to Decision Letter 0]

29 Oct 2022

Reviewer #1: 

Reviewer’s Comment:

Well written paper, with some minor English issues (see below)

Not very innovative, standar statistics and existing public software tools were used for evaluation of the effects of different methods for distorion correction. You could describe some the non-linearity models. No mathematical formalisms are presented, just mentioning and citing known analysis techniques.

Please enhance the quality of figures 2 and 3; the text is very blurred; I also recommend maximizing contrast by using black for the font.

Response:

Dear reviewer, thank you very much for your valuable feedback and suggestions regarding our manuscript. Indeed, our manuscript does not present mathematical formalisms or focusses too deeply on the technical details of non-linearity models or distortion corrections. This has been studied and published before by previous research groups in great detail which are cited in our manuscript. It was not the aim of our study to further elucidate the physical and technical details and characteristics of gradient non-linearities or mechanisms of distortion correction. In our experience, the application of distortion correction is very common in clinical routine imaging. However, in research publications there is no mentioning of wether it was applied or not. Therefore the main aim of our study was to raise awareness in the research community by showing the effect of distortion correction on a very practical research and clinical task, applying common techniques such as volumetric analysis using freesurfer. Our presented study focusses on the actual effect and extent which distortion corrections can have on volumetric analysis. With our study we would like to emphasize that in addition to all other common methodological details that are reported in studies or by commercial software to ensure reproducibility, which is essential in clinical routine and research, the application of distortion correction is a relevant methodological detail that might be a source of very different measurements. In our opinion it is essential to understand the magnitude of the influence on volumetric assessments whether distortion corrections have been applied to the images or not. We therefore used the standard software provided by the MRI vendor in combination with a frequently used tool for brain segmentation to create a typical research or clinical routine setting. Especially with the growing popularity of commercial segmentation and volumetric MRI tools our results should raise awareness of potential sources or error. To clarify this, we added following sentences to the Introduction and Discussion:

“The effect of correcting image distortions on volumetric measures has been described before by numerous studies in great technical detail [5-8]. Those previous studies focused on the feasibility of their in-house developed distortion correction methods and did not report the impact of gradient non-linearities on volumetric measures. With the increasing availability of commercial software, volumetric analysis will be applied more frequently in clinical routine and research and clinicians as well as researchers should be aware of confounding factors such as gradient nonlinearities. Thus, the aim of the present work was to quantify the effect and magnitude of distortion corrections by utilizing a typical and practical scenario, i.e. volume measurements using the distortion correction methods provided by the MRI vendor. We hypothesized that the application of distortion correction has a significant and relevant impact on volumetric measurements.”

“With the growing popularity of commercial volumetric tools volumetric analysis will be conducted on a more regular basis in clinical routine. Therefore, clinicians and researchers must be aware of the effects caused by distortions on volumetric results, especially when using cut-off values to differentiate between healthy aging and affected patients.”

We have enhanced the quality of figures 2 and 3. 

Reviewer’s Comment:

Page2, line 22

lateraloccipital and postcentral

lateral occipital and postcentral

THIS IS PREFERRED, "postcentral" is fine

Please make this correction in all the document

Response:

Thank you for pointing this out. We corrected this in the manuscript.

Reviewer’s Comment:

Page 15, line 18

Of note, it was reported by Jovicich

? what do you mean by "Of note"? ask help form an English.spoken person

Response:

This mistake was corrected and the sentence was rephrased as following:

“It was reported by Jovicich et al. that head positioning can have an impact on distortion effects [5].”

Additionally, the manuscript was carefully checked for further grammatical and semantic errors and corrected if necessary.

Reviewer’s Comment:

page 17 line 13

captured in or study

included in our study <- you refer to sources of distortion, the use of "capture" is incorrect.

Response:

We corrected the sentence accordingly.

“In addition, as stated above, there are other sources for distortions in brain MRI that can influence volumetric analysis but were not included in or study.”

Reviewer #2: 

Reviewer’s Comment:

This paper looks at the effect of gradient non-linearities corrections on 3D T1-w magnetic resonance (MR) images. Specifically, the paper compares the estimates from the output of the Freesurfer processing pipeline such as brain volumes and regional cortical thickness from images with and without correction of non-linearities (DC, as per authors). There is a lot of work in the paper, and I found the figures to be essential to the understanding of the manuscript.

Introduction

I found the introduction lacking fundamental details. The literature has significant contributions trying to address gradient non-linearities and their impact to the reconstructed images. They present the source, the physics behind their correction and potential improvements while their limitations. The authors site only three papers that address image distortions corrections. However, these only describe some of the problems including multi-site data and these are not included in the introduction. I recommend expanding to add a short review of others’ findings.

Response:

Thank you very much for this valuable comment. It has already been shown by previous studies, who also address the sources and underlying physics of gradient non-linearities in greater detail, that gradient non-linearities can cause shrinking or dilatation of certain brain areas. These studies are cited in the introduction, and we added the study of Caramanos et al. to provide the reader with an additional source for a more detailed explanation of gradient non-linearities. However, it has not been stated by previous studies whether the differences between images with and without distortion corrections are actually significant or relevant in clinical routine. The focus of our manuscript is to show that there are significant differences in volumetric measures depending on whether distortion corrections have been applied or not. Further we wanted to quantify those differences and show their magnitude. In our opinion this is from upmost importance and it is crucial for every researcher and clinicians to understand the effect and magnitude distortion corrections can have on volumetric measures. With our study we would like to emphasize that in addition to all other common methodological details that are reported in studies or by commercial software to ensure reproducibility, which is essential in clinical routine and research, the application of distortion correction is a relevant methodological detail that might be a source of very different measurements.

To emphasize this we added following sentences to the introduction:

“The effect of correcting image distortions on volumetric measures has been described before by numerous studies in great technical detail [5-8]. Those previous studies focused on the feasibility of their in-house developed distortion correction methods and did not report the impact of gradient non-linearities on volumetric measures. With the increasing availability of commercial software, volumetric analysis will be applied more frequently in clinical routine and research and clinicians as well as researchers should be aware of confounding factors such as gradient nonlinearities. Thus, the aim of the present work was to quantify the effect and magnitude of distortion corrections by utilizing a typical and practical scenario, i.e. volume measurements using the distortion correction methods provided by the MRI vendor. We hypothesized that the application of distortion correction has a significant and relevant impact on volumetric measurements.”

Reviewer’s Comment:

The authors state that numerous factors can affect the images. A little naming of them is needed in the introduction. Furthermore, the authors indicate that these factors can be easily adjusted in a single site study. I don’t believe the statement is true. There are factors that are not related to the technology used, unavoidable changes to the technology, errors from the coils, factors associated with the participant, the physiology (such as flow, susceptibility), and positioning in the scanner. What are the inherent errors of perfect data? Multiple of these factors are not able to be corrected. Rudrapatna et al. demonstrated that motion influences the non-linearities corrections. I believe some of these should be introduce in the introduction

Response:

We totally agree with the reviewer’s statement. There are numerous factors than can affect images and only some of them can be corrected. To clarify this we added following to the introduction:

“Numerous technical factors can influence the results of brain volumetric analysis, such as acquisition protocols, intra- and interscanner variablilty, scanner system upgrades or movement [2, 3]. However, only some of these factors can be adjusted in single site studies, e.g. by using the same MR scanner, MR image sequence parameters, and software versions for the automatic brain volume analysis.”

Reviewer’s Comment:

The paper tests the use versus the non-use of a motion correction technique. However, which of these two approaches is correct or more accurate is not known. There is no ground true when scanning humans. Have the authors considered using phantom and what are the limitations of such approach? What is the novelty of this technique compared to others. There are multiple correction techniques, what differentiates this DC algorithm from others? There is a need to explain the correction technique, the expectations of the correction and the therefore elaborate a hypothesis. What problem is this paper really targeting? The hypothesis should be quantifiable and should state the reasoning to the change of using the DC technique. A significant change does not really indicate that there is a cause-consequence that drives a hypothesis.

Response:

Thank you for these interesting and insightful comments. We agree with the reviewer’s comment that it is always difficult to determine the ground truth in in vivo experiments, such as this presented research. However, it has been shown in previous studies and experiments that distortion corrections can alternate volumetric measures. These studies also used phantoms to validate their results and to explain the underlying physics of gradient nonlinearities. Therefore, the aim of our research was not to elucidate the mechanisms of distortion corrections or validate different algorithms, which has been demonstrated before. Our aim was to determine the impact and magnitude distortion corrections have on volumetric measures. We believe that our results will raise further awareness of the influences of distortion corrections, which is rarely reported in volumetric studies. The appeal of our research is to encourage researches to always state whether distortion correction was applied before reporting volumetric analysis, especially since there is no ground truth. To clarify our intentions and our hypothesis we rephrased the last paragraph of our introduction:

“The effect of correcting image distortions on volumetric measures has been described before by numerous studies in great technical detail [5-8]. Those previous studies focused on the feasibility of their in-house developed distortion correction methods and did not report the impact of gradient non-linearities on volumetric measures. With the increasing availability of commercial software volumetric analysis will be applied more frequently in clinical routine and clinicians should be aware of confounding factors such as gradient nonlinearities. Thus, the aim of the present work was to quantify the effect and magnitude of distortion corrections on volume measurements using the distortion correction methods provided by the MRI vendor. We hypothesized that the application of distortion correction has a significant impact on volumetric measurements.” 

Reviewer’s Comment:

Materials

Brain morphology is dependent on the age and gender of the participants. The demographics of this study are highly variable. A more detailed information will help understand the changes seen in the results. The study is based on one scanner, one software version, one protocol version, one correction technique. How long did it take to get the 36 subjects scanned? There are intricate changes to the technology that can biased the results here presented. Did anything else change? More detailed information is needed.

Response:

Numerous factors can have confounding effects on the quantification of brain volume, such as age, sex, hydration state, and lifestyle and risk factors. The reviewer is correct when stating that the demographics in our study are variable. However, we do not think that this is a meaningful problem when interpreting our results since comparison between volumetric measures of images with and without distortion corrected was performed patient-wise.

The images were acquired between March 2013 and February 2015. All images were acquired on the same scanner, using the same acquisition protocol and correction algorithm. Additional information was added to the manuscript.

Reviewer’s Comment:

The methods sections should have enough information to allow others to reproduce the experiment. This is a requirement for transparency and reproducibility of any scientific publication. I think there are many sections that can be improved, and more information is needed.

• On the DC technique to further assess the results. This include the algorithm itself and the software version and others.

Response:

The distortion correction technique was implemented in the software as provided by the vendor (syngo MR E11, Siemens Medical Solutions, Erlangen, Germany). We used the standard algorithm, which we stated in our methods. By using the standard algorithm supplied by the vendor we believe that our methods and results are highly comprehensible to other researchers. 

Reviewer’s Comment:

• The acquisition process, not just only the T1-w. length to acquire 36 participants, coil, landmarking, motion correction, etc.

Response:

A 32 channel head coil was used and each patient was imaged with full brain coverage. No additional motion correction was applied. We added these important details to the manuscript. 

Reviewer’s Comment:

• Freesurfer v7.1 is known to provide inaccurate parcellations of the brain, compared to previous versions. It often overestimates the temporal and parietal cortex. This refers to the robustness of the Freesurfer pipeline. Have the authors encounter these issues? Was there any mitigation action to correct the template transformations? Did the pipeline behave similarly between the nDC and the DC corrected data? Did all the transformations work accurately? How was this evaluated?

Response:

Thank you for this interesting comment. As the reviewer has stated before it is difficult to determine a ground truth in imaging studies. Therefore, we cannot say whether overestimation of the temporal and parietal cortex occurred in our results, especially since we did not investigate this occurrence in particular. Each Freesurfer segmentation and parcellation was inspected visually by an experienced neuroradiologist with 8 years of experience. We added this information to our manuscript. However, the same algorithm was used for both images, with and without distortion correction. So the algorithm should have the same effect on both images. For this analysis, we were not interested in providing a solution or correction for distortion artifacts. The main purpose of this study was to raise awareness among researchers that it does make a difference whether they use one image or the other and that the difference is relevant. We therefore advocate to encourage authors to describe these technical details in future studies.

Reviewer’s Comment:

• The estimate of the isocentre needs an explanation. What was the purpose or intent for its estimate. The paragraph as listed does not indicate the intention of the estimate use.

Response:

Thank you for this remark.

The isocentre was estimated applying the corresponding FreeFurfer utilities based on the information stored in the DICOM header and then used to calculate the distance for each freesurfer parcel from the isocenter. As technical studies have shown that distortion artifacts are lower in the isocenter and increase with distance, we hypothesized that this might also be measurable in our data. We added a sentence to the hypotheses and additional information to the methods, accordingly. 

Reviewer’s Comment:

• CCC is considered similar to ICC. How was it calculated, there are multiple ICC estimates depending on the data (see Koo 2016). When reporting ICC, the confidence intervals should also be reported. These determine if the ICC is excellent, good, or poor.

Response:

To address this comment we added a Supplemental Table reporting the confidence intervals for the CCC.

Reviewer’s Comment:

• LME model: given that the hypothesis is measuring difference, I understand the model used to fit the difference between the 2 images. However, where all the fixed effect used in the model simultaneously or one at the time? The explanation needs more information on the purpose of the modeling. The LME was not used to model the data, but it was used to measure factors that could influence the difference in the estimates. Results indicate a somewhow different use of the model. How was the p -value estimated? Was it a simple ANOVA between 2 different model? Which factors were used and why? If you use volume and thickness, are you expecting edge voxels to influence the results? In addition, given the variability in the demographics of the cohort used in the study, why were age and gender not used as factors as well?

Response:

The fixed effects in our model were all used simultaneously. We used the function lmer in R to create the model and the model was used to investigate factors that could influence the difference of the estimates. The p values were determined by using ANOVA. However, we understand that determining p values can be challenging for linear mixed models and it is often recommended to omit them. We added this to the limitations of this paper. 

We did not use age and gender as factors since we already used patient ID as random effect in our model. 

Results:

Reviewer’s Comment:

Initially I found the results interesting and was surprised to see the region dependency of the estimates. It can be expected from the algorithm, and its special dependency in the correction technique. However, physiological variability can also lead to distortions that the algorithm can either not be able to or inaccurate try to correct. This is dependent of the DC algorithm.

As I looked further, I found some confusing information

• I recognized this is a small cohort. There is no information on how the difference, or the estimates were distributed across participants. Are these normally distributed? If not, is the t-test and LME model appropriate or is there a need for non-parametric test?

• Figure 2 and 3: they report % change. Percentage normalized to which value? I think the formula of the difference % [nDC-DC]/??? should be present in the axis to help their interpretation. This normalization can also bias the results. It is currently sorted based in most negative change to most positive change. When sorted based on anatomical region, were other information that could be inferred? Like most regions in the frontal lobe have statistically significant changes based on the t-test and why? What drives the changes in some regions compared to others.

Response:

The percentage difference refers to the difference between the nDC and the DC images. The formula would be (DC(thickness)-nDC(thickness)/nDC(thickness). We will add this formula to the figure caption. Secondly, it is highly interesting to understand what drives the changes in some regions compared to others. As we speculate in the Discussion a possible explanation for this finding might be that gradient linearity decreases significantly with increasing distance from the magnetic isocenter, which is pronounced along the z-axis. Our findings support this with distance to the magnetic isocenter along the z-axis as positive predictor for thickness and volume differences between the DC and nDC images.

Reviewer’s Comment:

• Table 1: If the 95% CI for the difference and therefore associated with Figue 2 and 3? For example, Caudalmiddlefrontal, it is not statistically significance different (t-test) in the volume, however, the variability in the difference estimates is high and crosses zero. Are there outliers, a particular participant driving this variability, is it a cohort variability? Further explanation is needed. The text speaks of summary values, which are usually good as a start point, but needs further detailed description of some examples.

Response:

Please be careful when interpreting the displayed 95% CI in Table 1 and Fig 2 and 3. Table 1 displays the absolute values for each cortical ROI while Figure 2 and 3 refer to the percentage changes between the nDC and DC images. 

Reviewer’s Comment:

• Figure 4, what drive the low CCC values? Is it the within variability of the measures?

Response:

Thank you very much for this interesting comment. We do not think that the low CCC values in some cortical ROIs is caused due to the within variability of the measures, since most of the cortical ROIs display high CCC values. Low CCC values rather seem to be location depending since most ROI with low CCC value are located frontal and parietal. 

Reviewer’s Comment:

Discussion

The discussion section describes the need to report the type of data (nDC or DC) used in the estimates to infer changes. I agree with the authors that this is relevant and important and should always be described in a paper. Again, this aligns with the importance of transparency and reproducibility of published data. However, in cross-sectional, particularly at a single centre, if all data are treated in the same manner, the biases due to non-linear gradients can be considered similar to all participants if collected within a short time frame. Furthermore, you might not be affected by the correction, considering that this is not better than non-corrected data, if not applied at all. However, the biases due to anatomical variability and others are still present. In longitudinal data there are other sources of errors that can affect your measure more than non-linear gradients.

The authors highlight the importance of the changed seen and compared those to anatomical changes over time in different pathologies reported in the literature. In the review, caution should be considered and group results by similarly managed data. Each study had chosen a processing pipeline and sometimes papers cannot be compared even less grouped together. I agree with the authors that head positioning is an important and unavoidable factor. How can the effect of this be minimized?

I find it difficult to relate the discussion to the proposed hypothesis. Can the hypothesis really be proven if not all regions are affected similarly? Are there edge-effects or outliers in the ROIs used? If so, how was this evaluated and corrected for?

How does the physiology affect the DC algorithm?

Response:

Thank you very much for this detailed comment. As described by the reviewer the focus of this presented study was to demonstrate the magnitude of distortion corrections on volumetric measures employing a practical example and raise awareness for the existence, magnitude and comparability of this effect on volumetric analysis. Of course, there are plenty of other technical and physiological factors that can influence volumetric analysis. Though many of them cannot be corrected, there are existing algorithms to reduce the influence of gradient non-linearities. The awareness of this seems to be even more necessary today with growing popularity of commercial volumetric tools and more frequent use in clinical routine. To address this topic, we added following sentences to the discussion:

“With the growing popularity of commercial volumetric tools volumetric analysis will be conducted on a more regular basis in clinical routine. Therefore, clinicians and researchers must be aware of the effects caused by distortions on volumetric results, especially when using cut-off values to differentiate between healthy aging and affected patients.”

Small displacements of the patient’s head have a notable effect on the volumetric measures. This has been described before in more detail (Caramanos Z, et al. 2010). Positioning of the patient identically across scanning sessions can minimize this effect, but this is very difficult. A better solution is to apply distortion corrections, as demonstrated in this study.

To address the reviewer’s concern regarding our hypothesis, we rephrased it accordingly:

“…, the aim of the present work was to quantify the effect and magnitude of distortion corrections on volume measurements using the distortion correction methods provided by the MRI vendor. We hypothesized that the application of distortion correction has a significant impact on volumetric measurements.”

Reviewer’s Comment:

• Use of acronyms such as MR and MRI, while known, should always be introduced. Same for DC.

Response:

The abbreviations are spelled out the first time they appear in the manuscript.

Reviewer’s Comment:

• Introduction: line 8 should be single site or single centre, and not single-side.

Response:

Done.

Reviewer’s Comment:

• Introduction: Page 5 line 1, confusing, needs references and further explanation.

Response:

The whole paragraph was re-written and the sentence was rephrased as well. 

Reviewer’s Comment:

• Figure1: please overlay with some transparency, hard to see the differences.

Response:

We revised Figure1 and added transparency to the overlays.

Reviewer’s Comment:

• Discussion: page 14 line 09 I would use dementia, as it is a spectrum. Also, it is Parkinson’s disease and there is a full stop after it that should be a comma.

Response:

Corrected.

Reviewer’s Comment:

• Discussion: page 14 line 22: This can lead… is repeated from 2 sentences before.

Response:

The sentence was removed from the manuscript.

Reviewer’s Comment:

• Discussion: page 15 line 22 should say cortical areas, not areals

Response:

Done.

---

## [Decision Letter · Decision Letter 1]

15 Feb 2023

PONE-D-22-16057R1Effect of Geometric Distortion Correction on Thickness and Volume Measurements of Cortical Parcellations in 3D T1w Gradient Echo SequencesPLOS ONE

Dear Dr. Thaler,

Thank you for submitting your manuscript to PLOS ONE. After careful consideration, we feel that it has merit but does not fully meet PLOS ONE’s publication criteria as it currently stands. Therefore, we invite you to submit a revised version of the manuscript that addresses the points raised during the review process. In particular, some important clarifications in the methods were requested by the reviewer. 

We look forward to receiving your revised manuscript.

Kind regards,

Dzung Pham

Academic Editor

PLOS ONE

Reviewers' comments:

Reviewer's Responses to Questions

**Comments to the Author**

1. If the authors have adequately addressed your comments raised in a previous round of review and you feel that this manuscript is now acceptable for publication, you may indicate that here to bypass the “Comments to the Author” section, enter your conflict of interest statement in the “Confidential to Editor” section, and submit your "Accept" recommendation.

Reviewer #3: (No Response)

2. Is the manuscript technically sound, and do the data support the conclusions?

Reviewer #3: Yes

3. Has the statistical analysis been performed appropriately and rigorously? 

Reviewer #3: Yes

4. Have the authors made all data underlying the findings in their manuscript fully available?

Reviewer #3: Yes

5. Is the manuscript presented in an intelligible fashion and written in standard English?

Reviewer #3: Yes

6. Review Comments to the Author

Reviewer #3: This paper investigates the effects of distortion correction on thickness and volume measurements from 3D T1w MR images using FreeSurfer and a commercially available distortion correction algorithm. Adequate data were acquired and processed to support the objective of this study. Careful proofreading, more elaboration, and better presentation of the results are needed.

***Introduction

In the sentence “…scanner system upgrades or movement”, what movement? Scanner movement?

***Materials and Methods

"No additional motion correction was applied."  "No motion correction was applied."

More details about the distortion correction algorithm are needed. Siemens provides two options (2D or 3D) for distortion correction. Which one was used? A brief description of the correction algorithm, which can be obtained from Siemens scientists/staff, will be helpful for readers to be aware of the differences between readers’ algorithms and the algorithm in this work.

Were any filters used?

*FreeSurfer cortical thickness and volume

Figure 1: It is very hard to visualize blue and purple. While it is good to inspect the whole brain, having close-up or zoomed-in views is needed to appreciate the differences between DC and nDC. For example, see figure 4 in https://pubmed.ncbi.nlm.nih.gov/31666530/. Showing the ROIs with most pronounced differences in thickness and volume measurements would be helpful. For example, precentral and paracentral ROIs.

Use either isocenter or isocentre.

It seems that the last paragraph in this section is to explain how the distance was calculated. In sentence “Isocenter coordinates for each patient were derived from the origin of the anatomical coordinate system that were transferred to the respective image coordinate system”, what is the isocenter in this sentence? Is it the magnetic isocenter mentioned in Statistics?

In “The isocentre … then used to calculate the distance for each FreeSurfer parcel from the isocenter. “, what are the two isocenters? Were they the magnetic isocenter that was mentioned later. Please clarify.

How was the distance calculated?

***Results

*Cortical thickness and volume

Since Table 1 is a busy table, adding horizontal gridlines will improve its readability. Further, it may be helpful to add a column on the left to assign a number to each ROI/row. Then use the numbers as y labels in Figures 2 and 3, whose y labels in the current format are too busy/blurry to read. Doing so will greatly improve the visualization of the figures. More comments about Figs 2 and 3 below.

In the title of Table 1:

The sentence “A p-value <0.05 was considered statistically significant” appeared in Statistics already and thus is redundant. It may be better to revise the last three sentences as two: “Significant differences (p < 0.05) are highlighted in bold. The false discovery rate (FDR) was used to correct for multiple comparisons.”

In Figures 2 and 3, y labels are too blurry to read. The information of the two figures is similar to/derived from Table 1. While it is nice to visualize the ranking of ROIs in terms of differences, it is more relevant to visualize the dependence of difference on distance, which is directly related to geometric distortion. Therefore, I suggest to add the following figures: mean-percentage-difference vs distance plots for thickness and volume measurements. Include 3 curves in each figure: mean-percentage-difference vs x, y, and z.

There is a formula/equation in the caption of Figures 2 and 3. It can be moved to be and described better in the section “FreeSurfer cortical thickness and volume”.

In Figure 4, use “Concordance correlation coefficient” is more descriptive/explicit than “CCC”. X labels are blurry.

***Discussion

“… were not included in or study”  “… were not included in our study”

“For the mixed linear model”  “For the linear mixed model”

To calculate p values for linear mixed model, please check “Evaluating significance in linear mixed-effects models in R” (https://link.springer.com/article/10.3758/s13428-016-0809-y).

7. PLOS authors have the option to publish the peer review history of their article (what does this mean?). If published, this will include your full peer review and any attached files.

Reviewer #3: No

---

## [Author Response · Author response to Decision Letter 1]

24 Feb 2023

Reviewer #3: This paper investigates the effects of distortion correction on thickness and volume measurements from 3D T1w MR images using FreeSurfer and a commercially available distortion correction algorithm. Adequate data were acquired and processed to support the objective of this study. Careful proofreading, more elaboration, and better presentation of the results are needed.

***Introduction

R1. In the sentence “…scanner system upgrades or movement”, what movement? Scanner movement?

A1. Thank you for pointing this out. Of course, patient movement was meant, and we corrected the sentence accordingly.

 “Numerous technical factors can influence the results of brain volumetric analysis, such as acquisition protocols, intra- and interscanner variablilty, scanner system upgrades or patient movement [2, 3].”

***Materials and Methods

R2. "No additional motion correction was applied."  "No motion correction was applied."

A2. The sentence was rephrased accordingly.

R3. More details about the distortion correction algorithm are needed. Siemens provides two options (2D or 3D) for distortion correction. Which one was used? A brief description of the correction algorithm, which can be obtained from Siemens scientists/staff, will be helpful for readers to be aware of the differences between readers’ algorithms and the algorithm in this work.

A3. In this study we used a 3D distortion correction. It is difficult to provide specific information about the algorithm since Siemens keeps its software confidential. However, we are able to refer to the patent of the distortion correction algorithm. (https://patents.google.com/patent/DE19540837A1/en)

R4. Were any filters used?

A4. No filters were applied to the acquired images.

*FreeSurfer cortical thickness and volume

R5. Figure 1: It is very hard to visualize blue and purple. While it is good to inspect the whole brain, having close-up or zoomed-in views is needed to appreciate the differences between DC and nDC. For example, see figure 4 in https://pubmed.ncbi.nlm.nih.gov/31666530/. Showing the ROIs with most pronounced differences in thickness and volume measurements would be helpful. For example, precentral and paracentral ROIs.

A5. We fully agree with the reviewer’s comment. It is difficult to visualize the subtle differences caused by distortion correction. For a better visualization, we added a new figure to the manuscript to display the effect of distortion correction. In this new figure, we also zoomed into the central region, where distortion effects were prominent. However, due to the subtle differences it is still difficult to visualize the effect on cortical thickness and volume.

R6. Use either isocenter or isocentre.

A6. Thank you for pointing this out. It was changed to isocenter.

R7. It seems that the last paragraph in this section is to explain how the distance was calculated. In sentence “Isocenter coordinates for each patient were derived from the origin of the anatomical coordinate system that were transferred to the respective image coordinate system”, what is the isocenter in this sentence? Is it the magnetic isocenter mentioned in Statistics?

In “The isocentre … then used to calculate the distance for each FreeSurfer parcel from the isocenter. “, what are the two isocenters? Were they the magnetic isocenter that was mentioned later. Please clarify.

How was the distance calculated?

A7. The reviewer’s assumption is correct. Every time the isocenter is mentioned, we refer to the magnetic isocenter. The coordinates of the magnetic isocenter can be obtained from the origin of the anatomical coordinate system. However, by using the recon-all command the cortical parcellation will be created in a new “FreeSurfer” space and not in the original T1 space and therefore it is necessary relocate the magnetic isocenter in the FreeSurfer space. Further, we calculated the distance between the magnetic isocenter, now transformed in the FreeSurfer space, and the center of gravity of each ROI.

For a better understanding we rephrased the paragraph and added additional information about the determination of the magnetic isocenter in the FreeSurfer space.

***Results

*Cortical thickness and volume

R8. Since Table 1 is a busy table, adding horizontal gridlines will improve its readability. Further, it may be helpful to add a column on the left to assign a number to each ROI/row. Then use the numbers as y labels in Figures 2 and 3, whose y labels in the current format are too busy/blurry to read. Doing so will greatly improve the visualization of the figures. More comments about Figs 2 and 3 below.

A8. Thank you very much for this excellent advice. We gladly followed the reviewer’s suggestion by adding horizontal gridline to the table. Also, we added a column with number from 1-31 assigned to the ROI labels.

Also, we added those labels to Figs 2 and 3 to improve the visualization.

R9. In the title of Table 1:

The sentence “A p-value <0.05 was considered statistically significant” appeared in Statistics already and thus is redundant. It may be better to revise the last three sentences as two: “Significant differences (p < 0.05) are highlighted in bold. The false discovery rate (FDR) was used to correct for multiple comparisons.”

A9. We changed the sentence accordingly.

R10. In Figures 2 and 3, y labels are too blurry to read. The information of the two figures is similar to/derived from Table 1. While it is nice to visualize the ranking of ROIs in terms of differences, it is more relevant to visualize the dependence of difference on distance, which is directly related to geometric distortion. Therefore, I suggest to add the following figures: mean-percentage-difference vs distance plots for thickness and volume measurements. Include 3 curves in each figure: mean-percentage-difference vs x, y, and z.

A10. Thank you very much for this comment. We added two new figures to the manuscript as suggested by the reviewer. 

R11. There is a formula/equation in the caption of Figures 2 and 3. It can be moved to be and described better in the section “FreeSurfer cortical thickness and volume”.

A11. We moved the equation to the section “FreeSurfer cortical thickness and volume”

R12. In Figure 4, use “Concordance correlation coefficient” is more descriptive/explicit than “CCC”. X labels are blurry.

A12. We changed the heading of the figure and improved the image’s resolution.

***Discussion

R13. “… were not included in or study”  “… were not included in our study”

R14. “For the mixed linear model”  “For the linear mixed model”

A13 and A14. Thank you for pointing out these typos. We corrected them in the manuscript accordingly.

R15. To calculate p values for linear mixed model, please check “Evaluating significance in linear mixed-effects models in R” (https://link.springer.com/article/10.3758/s13428-016-0809-y).

A15. We used the function lmer in R to create the model and the model was used to investigate factors that could influence the difference of the estimates. The p values were determined by using ANOVA. However, we understand that determining p values can be challenging for linear mixed models and it is often recommended to omit them. We added this to the limitations of this paper.

---

## [Decision Letter · Decision Letter 2]

28 Mar 2023

PONE-D-22-16057R2Effect of Geometric Distortion Correction on Thickness and Volume Measurements of Cortical Parcellations in 3D T1w Gradient Echo SequencesPLOS ONE

Dear Dr. Thaler,

Thank you for submitting your manuscript to PLOS ONE. After careful consideration, we feel that it has merit but does not fully meet PLOS ONE’s publication criteria as it currently stands. Therefore, we invite you to submit a revised version of the manuscript that addresses the points raised during the review process. Please clearly indicate the distortion correction setting within the manuscript in the Materials and Methods section. To verify this setting on Siemens dicom files, this can usually be found in the Image Type dicom tag (0008,0008) and is indicated by ND for no distortion correction, DIS2D for 2D distortion correction, or DIS3D for 3D distortion correction. Alternatively, packages like GDCM can also be used to examine the Siemens private header under "sDistortionCorrFilter.ucMode". A value of 1 indicates no distortion correction, 2 indicates 2D, 4 indicates 3D. This setting is important to fully understand the implications of this study (cf. https://www.ncbi.nlm.nih.gov/pmc/articles/PMC8971027 and  https://pubmed.ncbi.nlm.nih.gov/16757858/).

We look forward to receiving your revised manuscript.

Kind regards,

Dzung Pham

Academic Editor

PLOS ONE

Journal Requirements:

Reviewers' comments:

Reviewer's Responses to Questions

**Comments to the Author**

1. If the authors have adequately addressed your comments raised in a previous round of review and you feel that this manuscript is now acceptable for publication, you may indicate that here to bypass the “Comments to the Author” section, enter your conflict of interest statement in the “Confidential to Editor” section, and submit your "Accept" recommendation.

Reviewer #3: (No Response)

2. Is the manuscript technically sound, and do the data support the conclusions?

Reviewer #3: Yes

3. Has the statistical analysis been performed appropriately and rigorously? 

Reviewer #3: Yes

4. Have the authors made all data underlying the findings in their manuscript fully available?

Reviewer #3: Yes

5. Is the manuscript presented in an intelligible fashion and written in standard English?

Reviewer #3: Yes

6. Review Comments to the Author

Reviewer #3: acquired within 5m55s -> acquired in 5m55s

As mentioned previously, please specify 2D or 3D correction used for the study. And discuss/mention possible differences between the 2D and 3D corrections or lack of comparing the two may be a limitation of this study in discussion/limitations.

It is nice to compare images with and without geometry correction in Figs 1A-D.

In Figs 1C and D, if possible please add arrows to point out the regions/edges that show the effects of the correction.

Labels E and F in Fig 1 are missing.

As pointed out by the statement "The distortion effects were most prominent in the central region but difficult to detect visually" in Fig 1 caption, is it possible to have close-up views of an off-isocenter region of E and F to show purple and blue regions better?

For Fig 4, my previous suggestion of using label IDs does not seem to be much helpful. In order to improve the readability of this figure, please consider to move the ID/label to close to left and right alternately of the error bars. Doing so, a larger font size can be used. e.g. following is a cartoon version of the aforementioned arrangement:

4.Entorhinal |---------O---------|

|-------------------------O-----------------------------| 8.Insula

And use "Cortical ROIs" as the label for y axis.

In Fig 5, all labels should be in black, instead of gray. These labels are a bit blurry. Font size of X labels need to be increased to be readable.

Supplemental Fig 1 and 2 show that percentage differences in cortical thickness and volume are minimal at a location ~(125,125,125) mm away from the scanner isocenter. This is counter-intuitive. At isocenter, gradient is 0 and B0 is more uniform, shouldn't geometric distortion be less influential? Please clarify.

7. PLOS authors have the option to publish the peer review history of their article (what does this mean?). If published, this will include your full peer review and any attached files.

Reviewer #3: No

---

## [Author Response · Author response to Decision Letter 2]

29 Mar 2023

Reviewer #3: 

R1. acquired within 5m55s -> acquired in 5m55s

A1. We changed the sentence accordingly.

R2. As mentioned previously, please specify 2D or 3D correction used for the study. And discuss/mention possible differences between the 2D and 3D corrections or lack of comparing the two may be a limitation of this study in discussion/limitations.

A2. We added the information to the manuscript that the distortion correction used in this study is a 3D correction. We agree with the reviewer that we cannot compare our results with 2D correction methods which are still used for distortion corrections. To further address this we added following sentences and reference to our limitations.

“Also, we used a 3D correction method and cannot our results with studies using a 2D correction method which are still in use by many vendors. However, the advantage of 3D distortion correction methods is their ability to provide more accurate and precise correction of image distortions, leading to improved image quality, better diagnostic accuracy.[1]”

R3. It is nice to compare images with and without geometry correction in Figs 1A-D.

In Figs 1C and D, if possible please add arrows to point out the regions/edges that show the effects of the correction.

Labels E and F in Fig 1 are missing.

As pointed out by the statement "The distortion effects were most prominent in the central region but difficult to detect visually" in Fig 1 caption, is it possible to have close-up views of an off-isocenter region of E and F to show purple and blue regions better?

A3. We understand the reviewer’s suggestions and tried to display the differences caused by the distortions on 3D T1w images as clearly as possible. Due to the subtle differences it is difficult to clearly demonstrate the distortions in the images. However, we believe that Fig 1 and 2 display the distortion effects in T1w images as accurate as possible. We added the labels E and F in Fig 1. 

R4.For Fig 4, my previous suggestion of using label IDs does not seem to be much helpful. In order to improve the readability of this figure, please consider to move the ID/label to close to left and right alternately of the error bars. Doing so, a larger font size can be used. e.g. following is a cartoon version of the aforementioned arrangement:

4.Entorhinal |---------O---------|

|-------------------------O-----------------------------| 8.Insula

And use "Cortical ROIs" as the label for y axis.

A4. Thank you for this suggestion. We changed Figure 3 and 4 accordingly.

R5. In Fig 5, all labels should be in black, instead of gray. These labels are a bit blurry. Font size of X labels need to be increased to be readable.

A5. We changed the colour and font size of the labels.

R6. Supplemental Fig 1 and 2 show that percentage differences in cortical thickness and volume are minimal at a location ~(125,125,125) mm away from the scanner isocenter. This is counter-intuitive. At isocenter, gradient is 0 and B0 is more uniform, shouldn't geometric distortion be less influential? Please clarify.

A6. 

We agree with the reviewer that this is counter-intuitive and also not correct. When creating Supplemental Fig 1 and 2 we picked the wrong column for distance from the isocenter. We recreated both figures with the correct data. Now, minimal differences are seen in proximity to the magnetic isocen

---

## [Editor Report · Decision Letter 3]

3 Apr 2023

Effect of Geometric Distortion Correction on Thickness and Volume Measurements of Cortical Parcellations in 3D T1w Gradient Echo Sequences

PONE-D-22-16057R3

Dear Dr. Thaler,

We’re pleased to inform you that your manuscript has been judged scientifically suitable for publication and will be formally accepted for publication once it meets all outstanding technical requirements.

Kind regards,

Dzung Pham

Academic Editor

PLOS ONE
---

## [Editor Report · Acceptance letter]

6 Apr 2023

PONE-D-22-16057R3 

Effect of Geometric Distortion Correction on Thickness and Volume Measurements of Cortical Parcellations in 3D T1w Gradient Echo Sequences 

Dear Dr. Thaler:

I'm pleased to inform you that your manuscript has been deemed suitable for publication in PLOS ONE. Congratulations! Your manuscript is now with our production department. 

Kind regards, 

on behalf of

Dr Dzung Pham 

Academic Editor

PLOS ONE